# Genome-Wide Identification of the ARF Gene Family in Safflower (*Carthamus tinctorius* L.) and Their Response Patterns to Exogenous Hormone Treatments

**DOI:** 10.3390/ijms26083773

**Published:** 2025-04-16

**Authors:** Shuwei Qin, Xinrong Wen, Mengyuan Ma, Jiaxing Wang, Jianhang Zhang, Meihui Huang, Kexin Sun, Ya Zhao, Meng Zhao, Asigul Ismayil, Min Liu, Aiping Cao

**Affiliations:** 1College of Life Sciences, Shihezi University, Shihezi 832003, China; qsw2623201516@163.com (S.Q.); 15299941807@163.com (X.W.); mengyuanma2022@163.com (M.M.); 18119249542@163.com (J.W.); zjh17630412656@163.com (J.Z.); 18503174612@163.com (M.H.); 13223072825@163.com (K.S.); 18160362098@163.com (Y.Z.); zm022362@163.com (M.Z.); asgli12@163.com (A.I.); 2Key Laboratory of Xinjiang Phytomedicine Resource and Utilization, Institute for Safflower Industry Research, Pharmacy College, Shihezi University, Ministry of Education, Shihezi 832003, China

**Keywords:** ARF, safflower, *Aux/IAA*, hormone treatment, active components, flower color

## Abstract

Auxin response factors (ARFs) are a class of transcription factors widely present in plants. As an important economic crop, research on the effects of safflower ARFs on endogenous auxin and effective components is relatively limited. In this study, a total of 23 ARF genes were identified from the safflower genome. Sequence alignment and domain analysis indicated the presence of conserved B3 and Auxin_resp domains in these ARFs. Phylogenetic analysis indicated that CtARF could be classified into five subfamilies, a conclusion also supported by gene structure, consensus motifs, and domain compositions. Transcriptome data showed that ARFs are expressed in all flower colors, but the expression levels of ARF family members vary among different flower colors. *CtARF19* had relatively higher expression in deep red flowers, *CtARF3* had higher expression in white flowers, *CtARF2/12* had higher expression in yellow flowers, and *CtARF21/22* had higher expression in light red flowers. Protein–protein interaction network analysis indicated that ARF family members (CtARF2/3/4/5/15/18/19/22) are located within the interaction network. Cis-acting element analysis suggested that *CtARF* genes may be regulated by hormone treatment (AuxRR-core) and abiotic stress, and the results of qRT-PCR also confirmed this. Additionally, the content of endogenous auxin and active components in safflower with different flower colors significantly changed upon treatment with hormones that affect IAA content. In summary, our study provides valuable insights into the biological functions of *CtARF* genes under exogenous hormone conditions and their effects on active components.

## 1. Introduction

Safflower (*Carthamus tinctorius* L.) has been cultivated in China for a period exceeding 2100 years. Its distribution is primarily concentrated in Xinjiang, Yunnan, Henan, Inner Mongolia, and several other regions. Remarkably, the cultivation area of safflower in Xinjiang constitutes over 80% of the total cultivation area across China. Moreover, safflower is cultivated in numerous regions across the globe, including countries such as the United States, India, and Canada. Safflower is a versatile economic crop, integrating medicinal [1], dyeing [2], oil-yielding [3], and feed [4] functions, and thus holds broad prospects for development and application. Hydroxysafflor yellow A (HSYA), extracted from the inflorescences of safflower, is composed of one C-glucosylquinochalcone and has been widely applied in the treatment of coronary heart disease, cerebral infarction, and tumors [5,6,7,8]. Therefore, investigating the regulatory mechanisms of safflower development and screening for potential genes that regulate HSYA biosynthesis are important approaches to improve the agronomic traits of safflower and increase the content of active components.

Auxin (*Aux/IAA*) is a class of plant hormones that plays a crucial role in plant growth and development [9]. It not only influences processes, such as plant embryo development and cell elongation, but also affects apical bud growth, root growth, and fruit maturation [10]. Due to the extensive functional repertoire of auxin, studying its biological functions through auxin-deficient mutants presents significant challenges. However, this difficulty can be overcome by employing auxin inhibitors to reduce auxin activity [11]. Different auxin inhibitors exert distinct inhibitory effects. For instance, N-1-naphthylphthalamic acid (NPA) is a chemical that alters IAA transport and thus affects the responses to indigenous IAA in planta. Scientifically modulating the ratio of these two in plants can regulate plant growth processes by affecting endogenous hormone levels, and can also achieve effective weed control in agricultural management [12].

The Aux/IAA and ARF families are core components of the auxin signaling pathway [13]. The Aux/IAA proteins can be divided into three functional domains: the N-terminal DNA-binding inhibitory domain, Domain II, and the C-terminal domain [14]. The DNA-binding inhibitory domain can recruit the TOPLESS (TPL) co-repressor, and the specific binding between them mediates the regulation of biological processes by auxin [15]. Domain II can interact with the TIR1/AFB co-receptor. The C-terminal domain, also known as the PB1 domain, consists of the Phox and Bem1 modules, which can be referred to as Domain III and Domain IV [16]. ARFs can also be divided into three functional domains: the DNA-binding domain (DBD), the middle region (MR), and the C-terminal dimerization domain (CTD) [17,18], which are involved in the regulation of gene activation or repression. The ARF family exhibits species-specific characteristics. Initially, 23 members of the ARF family were identified in *Arabidopsis thaliana*. Subsequently, 25, 23, 25, 36, and 24 members were identified in *Oryza sativa*, *Triticum aestivum*, *Sorghum bicolor*, *Setaria viridis*, and *Sesamum indicum*, respectively [19].

The ARF family is not only involved in the regulation of plant growth and development but also plays a crucial role in plant responses to abiotic stress and in mitigating heavy metal stress [20,21,22]. Previous studies have revealed that in *Arabidopsis thaliana*, *AtARF1* and *AtARF2* are involved in the regulation of flower organ development and senescence [23]. *AtARF6* and *AtARF8* promote the maturation of flower organs [24], while *AtARF17* facilitates anther development and pollen tube wall formation [25]. Concurrently, studies have demonstrated that ARF genes in apples can regulate anthocyanin biosynthesis pathways, thereby influencing flower color [26]. Therefore, ARF genes also possess potential regulatory functions in flower color formation. Flower color, as an important trait of plants, not only provides ornamental value, but also serves as a crucial signal to attract insects for pollination [27]. Previous studies have shown that the flower color of safflower is controlled by two dominant epistatic loci, each with two alleles, and these loci follow Mendelian inheritance laws [28,29]. Studies have shown that there are significant differences in HSYA content among different genotypes of safflower [30], and that HSYA content is highly correlated with flower color. Studies have indicated that genes encoding cinnamate-4-hydroxylase (C4H), chalcone synthase (CHS), C-glycosyltransferase (CGT), and cytochrome P450 (P450) may be involved in the biosynthetic pathway of HSYA [30]. Additionally, kaempferol (KAE) is synthesized through the catalytic actions of enzymes, such as cinnamate-4-hydroxylase (C4H), chalcone synthase (CHS), chalcone isomerase (CHI), and flavanone 3-hydroxylase (F3H). The biosynthetic pathways of KAE and HSYA indeed intersect, with the biosynthetic route of naringenin chalcone being conserved [30]. However, there is a lack of experimental evidence linking different flower colors of safflower to HSYA biosynthesis. Additionally, the mechanisms by which ARF family genes influence plant growth, development, and HSYA content in different flower colors are worth further investigation.

In this study, bioinformatics techniques were employed to identify *CtARF* genes from the whole genome of safflower. A comprehensive analysis of this gene family was conducted in terms of physicochemical properties, conserved sequences, gene structure, phylogenetics, and gene expression. Additionally, the effects of exogenous hormone treatment on safflower ARF genes, endogenous IAA, and active components were investigated. This study provides a theoretical basis for further exploring the functions of *CtARF* family genes and their involvement in the auxin pathway, which affects the growth, development, and synthesis of HSYA in safflower.

## 2. Results

### 2.1. Identification and Physicochemical Property Analysis of ARF Transcription Factor Family Members in Safflower

With the protein sequences of the ARF family in *Arabidopsis thaliana* and *Helianthus annuus* as references, 89 homologous genes (preliminary screen of the safflower genome) were obtained from the safflower genome through BLAST alignment. The identified proteins were successively submitted to the Pfam website for a domain search, and the presence of the NBD and CTD domains was confirmed. In summary, A total of 23 ARF genes were identified in the safflower genome, and they were named *CtARF1* to *CtARF23* according to their chromosomal locations (Appendix A). The protein lengths of CtARF family members ranged from 548 to 1119 aa (Appendix A), with molecular weights of 61.87–125.48 kD and aliphatic indices of 63.41–76.91. The isoelectric points of the encoded proteins of CtARF family genes ranged from 5.13 to 7.64, suggesting that they may function in weakly acidic or neutral subcellular environments. The instability indices of all encoded proteins of CtARF family genes were greater than 40, indicating that they are unstable proteins. The grand average of hydropathicity values of all proteins was negative, indicating that they are hydrophilic proteins. Subcellular localization prediction showed that all proteins are located in the nucleus (Table 1).

### 2.2. Chromosomal Distribution of ARF Family Members in Safflower

Based on the genome annotation information of safflower, the chromosomal localization of ARF genes was conducted (Appendix A). ARF genes were distributed across all 12 chromosomes of safflower. Specifically, chromosomes 2, 3, 4, 6, 7, and 9 each harbored one ARF gene, chromosomes 1, 10, and 12 each had two ARF genes, chromosomes 5 and 8 each possessed four ARF genes, and chromosome 11 contained three ARF genes. These findings indicated that ARF genes in safflower were dispersed throughout the chromosomal set.

### 2.3. Phylogenetic Analysis of ARF Proteins in Safflower

To further investigate the evolutionary relationships of *CtARF* genes and explore their potential functions, a phylogenetic tree was constructed using 85 members of the ARF family from safflower, *Arabidopsis thaliana*, and *Helianthus annuus* (Figure 1). Based on the phylogenetic tree and the classification of ARF genes in *Arabidopsis* and *Helianthus*, the *CtARF* family was divided into five subfamilies (I, II, III, IV, and V). The results indicated that safflower ARF genes shared similarities with those in *Arabidopsis* and *Helianthus*. Subfamily I included *CtARF3/4/10/16/19/22* (six members), subfamily II included *CtARF1/14* (two members), subfamily III included *CtARF6/7/8/9/15/18/23* (seven members), subfamily IV included *CtARF2/5/12/20* (four members), and subfamily V included *CtARF11/13/17/21* (four members).

### 2.4. Gene Structure Analysis of ARF Proteins in Safflower

The gene structure analysis of *CtARF* revealed that the ARF family genes in safflower contained 2–15 exons (Appendix A). Specifically, subfamily I genes had 12–15 exons, subfamily II genes had 14 exons, subfamily III genes had 8–14 exons, subfamily IV genes had 10–13 exons, and subfamily V genes had 2–5 exons. These results indicated that genes within the same subfamily were identical or similar, and the gene structure of subfamily V was significantly different from the other four subfamilies.

To elucidate the function and mechanism of ARF proteins in safflower, we conducted secondary and tertiary structure predictions of safflower ARF proteins. The secondary structure analysis revealed that the content of random coils in safflower ARF proteins was the highest, exceeding 60% in all proteins (except for ARF19, which had 59.67%). In contrast, the content of α-helices was over 10%, and the content of extended strands ranged from 8.77% to 14.96%. The β-sheet content was the lowest, around 2%. These results indicated that random coils are the predominant secondary structure component of ARF proteins and may play a role in the formation of the tertiary structure (Appendix A). We further modeled the tertiary structure of CtARF proteins, and the results showed successful modeling for all proteins (Appendix A). There were significant differences in the proportions of α-helices, extended strands, β-turns, and random coils among proteins from different subfamilies. The structures of CtARF4 and CtARF19 were highly similar, and ARF genes that clustered together in the phylogenetic tree encoded proteins with similar tertiary structures.

### 2.5. Analysis of Conserved Domains and Motifs in Safflower ARF Proteins

The analysis of conserved domains revealed that 13 safflower ARF proteins possessed B3, Auxin_resp, and AUX_IAA superfamily domains, including 6 members of subfamily I, 2 members of subfamily II, 4 members of subfamily III, and 1 member of subfamily IV. Additionally, 9 ARF proteins contained only B3 and Auxin_resp domains, including 2 members of subfamily III, 3 members of subfamily IV, and 4 members of subfamily V. Notably, CtARF18 in subfamily III lacked the B3 domain but contained Auxin_resp and AUX_IAA superfamily domains (Figure 2B). The conserved domains of safflower ARF proteins were predicted using the online software MEME (Version 5.5.7) (Figure 2C). The results showed that closely related members exhibited similar conserved motif characteristics. Subfamily I contained 8–10 conserved motifs, subfamily II contained 10 conserved motifs, subfamily III contained 6–10 conserved motifs, subfamily IV contained 7–10 conserved motifs, and subfamily V contained 8–10 conserved motifs. Specifically, CtARF2 had 7 conserved motifs, CtARF5/10/16/17 had 8 conserved motifs, CtARF3/11 had 9 conserved motifs, CtARF18 had 6 conserved motifs, and the other sequences all had 10 conserved motifs. All safflower ARF proteins possessed Motif6 and Motif9. These findings indicated that members of the same subfamily have identical or similar conserved structures, whereas CtARF18 was significantly different from the other sequences.

### 2.6. Analysis of Synteny and Interactions of Safflower ARF Proteins

Intragenomic synteny analysis revealed that there were evident duplication relationships of *CtARF* genes in the safflower genome. Among the 23 *CtARF* genes, 10 segmental duplication events were identified (Figure 3A), which were distributed across all chromosomes except chromosome 6. It is speculated that these genes may have similar functions. Synteny analysis between *Arabidopsis* and safflower (Figure 3C) revealed 14 pairs of homologous genes, indicating the conservation of ARF genes across different species.

To analyze the potential interactions among CtARF proteins, we compared the protein sequences of all ARF and IAA gene families in safflower using the STRING website to identify the highest-scoring homologous proteins, and the results were visualized using Cytoscape (Version 3.10.0) (Figure 3B). By aligning the ARF and IAA protein sequences with the STRING database, we identified a protein interaction network comprising 216 edges and 37 nodes. Among them, eight ARF family members (CtARF2/3/4/5/15/18/19/22) were located within the interaction network, indicating that they may be core members of the CtARF gene family regulating the growth and development of safflower. The remaining 29 members were located outside the interaction network, and some members were found to have no interactions. Notably, all core members were from the ARF family, suggesting that ARF family members have more diverse functions. Additionally, the GO enrichment results indicated that safflower ARF family members have a wide range of biological functions and primarily act as transcriptional activators to regulate gene expression (Appendix A).

### 2.7. Analysis of Cis-Acting Elements in the Promoter Regions of Safflower ARF Genes

The promoters (2000 bp upstream of the translation initiation codon) of the ARF family genes in safflower were analyzed (Appendix A). In addition to the core promoter elements (TATA-box) and upstream promoter elements (CAAT-box), 31 cis-acting elements were identified, which could be mainly classified into two categories: light-responsive elements and hormone-responsive elements (Appendix A). Among the 23 ARF genes, auxin-responsive elements (TGA-element and AuxRR-core) were detected, indicating that the ARF proteins likely played important roles in regulating IAA-related events. Moreover, nine types of light-responsive elements were found in the promoters of these genes, with each gene containing one or more light-responsive element. This suggests that light has a significant impact on the growth and development of safflower. Additionally, a variety of hormone-responsive elements, such as those for abscisic acid, methyl jasmonate, and auxin, were identified in the promoters of the ARF family genes, indicating that these genes may be involved in hormone regulation to cope with environmental changes. Further analysis revealed that different ARF members had distinct cis-acting elements (Appendix A). For example, *CtARF6* contained a large number of ABRE (10) and G-box (11) elements, suggesting its involvement in ABA regulation and light response pathways. *CtARF9* had a high number of CGTCA-motif (8) elements, which may be related to the expression of methyl jasmonate. *CtARF10* contained a large number of ARE (8) and 3-AF1 binding site (8) elements, which may be related to hormone expression, environmental stress, and transcriptional regulation. *CtARF12* and *CtARF20* contained a large number of ARE (9/8) elements, while *CtARF13* had a high number of ABRE (7), G-box (8), and Box II (8) elements. The ATCT-motif element was abundant in *CtARF15* and *CtARF17*. These results indicated that different genes have distinct cis-acting elements, and genes within the same subfamily share similar cis-acting elements.

### 2.8. Expression Analysis of Safflower ARF Genes in Different Flower Colors

Flower color, as a significant trait of safflower, not only provides ornamental value but also attracts pollinating insects. Hydroxysafflor yellow A (HSYA), composed of one C-glucosylquinochalcone and responsible for the yellow pigments in the red color of safflower, has been widely used in the treatment of diseases, such as coronary heart disease, hypertension, and cerebral thrombosis. The expression of the 23 ARF genes in different flower colors of safflower (deep red, light red, yellow, and white) was analyzed (Figure 4D). The transcriptome data were derived from the safflower (*Carthamus tinctorius* L.) transcriptome database constructed by Wang et al., with sequencing materials obtained from safflower inflorescences (Appendix A) [30]. Subsequently, the transcriptome data were analyzed using the “pheatmap” package in R to generate expression heatmaps. The results showed that all 23 ARF genes were expressed in different flower colors, with genes from the same subfamily exhibiting similar expression patterns. However, there were significant differences in the expression levels of ARF genes from different subfamilies across different flower colors. *CtARF7/8/23* had high expression in all flower colors, while *CtARF4/5/15/17/18* had relatively low expression in all flower colors. *CtARF19* had relatively higher expression in deep red flowers, *CtARF3* had higher expression in white flowers, *CtARF2/12* had higher expression in yellow flowers, and *CtARF21/22* had higher expression in light red flowers.

### 2.9. Content of Endogenous IAA and Active Components in Safflower with Different Flower Colors

The endogenous IAA content in safflower with different flower colors was measured (Figure 4A). The results showed that there was a significant difference in endogenous IAA content between RS (4.66 μg/g) and WS (4.22 μg/g). KAE is a flavonoid compound, whose biosynthetic pathway partially overlaps with that of HSYA, and it exhibits pharmacological functions, such as anticancer activity and antioxidant capacity. The content of two antioxidants, including HSYA and KAE, was also determined in safflower with different flower colors (Figure 4B,C). There were significant differences in the content of active components among safflower with different flower colors. The content of HSYA was higher in RS (2.29%) and lower in WS (0.37%), while the content of KAE was higher in WS (0.03%) and lower in RS (0.01%), showing an opposite trend.

### 2.10. Effects of Different Exogenous Plant Hormone Treatments on Agronomic Traits of Safflower

Safflower plants were treated with exogenous IAA and an inhibitor of IAA transport (NPA; Figure 5). For both red safflower (RS) and white safflower (WS), the plant height and stem diameter increased to some extent after IAA application (Figure 5F,H). With the increasing IAA concentration, the plant height and stem diameter first increased and then decreased. The highest plant height was observed with IAA75 treatment, with RS reaching 40.6 cm and WS reaching 20.73 cm. The maximum stem diameter was achieved at IAA50 for RS (0.69 cm) and IAA25 for WS (0.64 cm). Compared with CK, plant growth status improved after IAA application. However, the high IAA concentration (IAA100) showed some inhibitory effects compared with moderate IAA treatments. After NPA application, both RS and WS exhibited some degree of deformity, with more pronounced deformity in RS (Figure 5B,D). Specific manifestations included stem bending and leaf curling. The degree of malformation in RS was more severe, with a more pronounced degree of bending. Interestingly, plant height significantly increased with the increasing NPA concentration, reaching the maximum with NPA70 treatment, with values of 48.23 cm for RS and 28.8 cm for WS. Unlike plant height, stem diameter initially increased and then decreased with the increasing NPA concentration, reaching the minimum with NPA70 treatment, with values of 0.48 cm for RS and 0.49 cm for WS. The deformity also became more severe at higher NPA concentrations.

### 2.11. Expression Analysis of CtARF Genes and Determination of Endogenous IAA and Active Component Content After IAA and NPA Treatment

We selected treatments with more pronounced agronomic traits (IAA75 and NPA50) for subsequent experiments. By measuring the endogenous auxin content in safflower inflorescences after hormone treatment, we found that safflower with different flower colors exhibited different responses to exogenous hormones (Figure 6). Compared with CK, the endogenous IAA content in RS was highest when treated with NPA and then endogenous IAA, while WS exhibited the opposite trend. The auxin signaling pathway or auxin biosynthesis-related genes in RS may be more sensitive to exogenous IAA and NPA, thereby being activated and promoting the synthesis of endogenous auxin. In contrast, WS may be more sensitive to feedback regulation, and exogenous IAA treatment may inhibit the synthesis of endogenous hormones.

The expression of the ARF gene family in different organs of safflower was detected after spraying IAA and NPA (Figure 7 and Appendix A). The results showed that different ARF genes exhibited different expression patterns after exogenous application of auxin and auxin inhibitor. In the roots of RS, the expression trends of *ARF2/3/5/19/22* were consistent after exogenous IAA application, with increased expression levels, while NPA treatment reduced the expression levels significantly below those of CK. In contrast, the expression trends of *ARF4/15* were consistent, showing a downward trend compared with CK, and the expression levels after NPA treatment were significantly lower than those of CK and IAA treatments. In the roots of WS, a completely different trend was observed. Except for *ARF2*, the expression levels of the other family members were NPA > IAA > CK, with extremely significant differences between each treatment group. Compared with CK, the expression levels of the *ARF2* gene increased significantly after IAA and NPA treatments, but there was no significant difference between the two treatments. In the stems of RS, the expression trends of ARF gene family members were consistent, with significant increases after treatment, the highest increase being observed after IAA treatment. However, there were no significant differences in the expression levels of *ARF3/4/19/22* between IAA and NPA treatments. In the stems of WS, the responses of ARF family members to IAA and NPA were more complex. NPA treatment significantly reduced the expression levels of *ARF2/3*, while IAA treatment significantly increased the expression levels of *ARF4/19/22*. The expression trends of *ARF5* and *ARF15* were consistent, both showing a downward trend and reaching the lowest point after NPA treatment. In the leaves of RS, the expression levels of *ARF2/3/5/15/19/22* genes all increased, reaching the maximum after NPA treatment. The *ARF4* member showed a different trend, with significant increases after treatment, reaching the highest level after IAA treatment. Interestingly, in the leaves of WS, the expression trends of ARF family members were generally consistent. Compared with CK, gene expression levels showed some differences after IAA treatment, but the differences were not significant. However, a highly significant increase was observed after NPA treatment. In the flowers of RS, the expression trends of ARF family members were mainly divided into two patterns. Compared with CK, the expression levels of *ARF2/3/4/22* were NPA > IAA > CK, while those of *ARF5/15/19* were IAA > NPA > CK, with a highly significant increase after IAA treatment. In the flowers of WS, ARF family members exhibited different trends. The expression levels of *ARF2/3/4/5/15* gradually decreased and reached the lowest point after NPA treatment. In contrast, *ARF22* showed an opposite trend, with the highest expression level after NPA treatment.

HSYA is a flavonoid compound unique to the inflorescence of safflower. In safflower, the content of HSYA was relatively high, while the content of KAE was low. In contrast, in white flowers, the content of KAE was higher, whereas the content of HSYA was lower. Safflower with different flower colors exhibited distinct responses to treatments with IAA or NPA (Figure 8). For RS, the content of HSYA decreased to varying degrees after IAA treatment (except for a 10.88% increase in IAA100 and a 3.06% increase in NPA30). The most significant decreases were observed in the plants treated with IAA75 and NPA10, with reductions of 44.22% and 26.87%, respectively. In contrast to the changes in HSYA content, the content of KAE increased to some extent after exogenous hormone treatment. Compared with CK, IAA75 (150%) and IAA100 (150%) showed significant increases, as did NPA30 (154.99%) and NPA50 (150%). After treatment with high concentrations (IAA100 and NPA70), the HSYA content in WS was extremely low, at around 0.01%. Except for the NPA50 treatment, which showed a significant increase in HSYA content by 41.03%, all other treatments reduced the HSYA content in WS. The content of KAE in WS responded differently to different exogenous hormones. IAA treatment significantly increased the KAE content in WS, with the highest increase of 85% observed after WIAA75 treatment. In contrast, NPA treatment resulted in a significant decrease in KAE content in WS, with no significant differences in KAE content among different NPA concentrations.

## 3. Discussion

### 3.1. Identification and Structural Analysis of the CtARF

The ARF transcription factors play a crucial role in the auxin signaling pathway, thereby influencing plant growth and development [12,31,32,33]. Different ARF proteins can act as transcriptional activators or repressors, and whether the ARF functions as a transcriptional activator or repressor depends on the sequence and corresponding structure of the middle region of the protein [34,35]. In addition, auxins and auxin inhibitors can be used as plant growth regulators. Exogenous application can influence the endogenous hormone levels within plants, modulate the content of secondary metabolites, and thereby alter plant quality [12]. We performed a bioinformatics analysis of the whole genome of safflower and identified a total of 23 *CtARF* genes (Table 1). The results showed that these genes were dispersed across the 12 chromosomes of safflower, with chromosome 3 being the longest but having fewer ARF genes [36,37,38]. Through phylogenetic analysis, it was found that safflower, sunflower, and *Arabidopsis* are closely related in terms of evolution (Figure 1). The *CtARF* gene family could be divided into five subfamilies. Genes within the same subfamily were highly similar, but the gene structure of subfamily V differed significantly from the other four subfamilies, which may be related to their distinct functions. In this study, *CtARF12* and *CtARF13* were found to be tandemly duplicated (Figure 3). Collinearity analysis revealed ten segmental duplication events for *CtARF* genes, all of which were formed through random duplication events. Intergroup collinearity analysis identified 14 pairs of homologous genes, indicating the conservation of ARF genes across different species and their similar functions. From an evolutionary perspective, gene duplication leading to the expansion of gene families often results in genes with identical or similar structures and functions. However, in some cases, gene duplication events can also lead to the acquisition of new functions by homologous genes [39], thereby enhancing the adaptability of plants.

### 3.2. Structural Analysis of CtARF Protein

By predicting the secondary and tertiary structures of ARF proteins in safflower, it was found that the proportions of α-helices, extended strands, β-turns, and random coils in proteins of different subfamilies varied significantly (Appendix A and Appendix A). Among them, CtARF18 lacked the B3 domain compared with other sequences, while the ARF proteins in sunflower did not have this deficiency. The B3 domain is a component of the DNA-binding domain (DBD) at the N-terminus of ARF proteins. In *Arabidopsis thaliana* ARF1 and ARF5, it also includes a dimerization domain (DD), which is located on both sides of the B3 domain, as well as an auxiliary Tudor-like domain. The DD can mediate the homodimerization of AtARF1 and AtARF5, which is essential for the cooperative binding of AtARF5 to target DNA [40]. The absence of the B3 domain is also observed in other species, such as *Eucalyptus grandis* [41] and *Physcomitrella patens* [42]. Therefore, different ARF members have diverged during the evolutionary process and may also exhibit functional differences.

### 3.3. Cis-Acting Element and Interaction Network Analysis of CtARFs

By analyzing the 2000 bp promoter regions upstream of the ARF family genes, we identified not only the core promoter TATA-box and the upstream promoter element CAAT-box but also light-responsive and hormone-responsive cis-acting elements (Appendix A). Each *CtARF* gene contained light-responsive cis-acting elements, suggesting that *CtARF* genes may play important roles in light-signaling regulation. Additionally, auxin-responsive elements (TGA-element and AuxRR-core) were found upstream of *CtARF3*, *CtARF4*, *CtARF14*, *CtARF15*, *CtARF16*, *CtARF18*, *CtARF20*, *CtARF21*, and *CtARF23*, indicating that these genes may be involved in auxin signaling. The types and numbers of cis-acting elements in the upstream regions of different genes varied, which may confer specificity and diversity to the functions of these genes.

In the phylogenetic tree, *AtARF7* and *AtARF19*, which are located on the same branch, exhibit identical functions and are involved in auxin signal transduction in roots [43,44]. The genes *CtARF1* and *CtARF14* are located on the same branch in the phylogenetic tree and may also have similar functions. ARFs, as key auxin regulators, can interact with other biological factors to jointly regulate plant growth and development. Research has shown that *Arabidopsis* has a total of 29 Aux/IAA genes and 23 ARF genes [45]. *AtARF4* can regulate branch regeneration through synergistic action with *AtARF5* and *AtIAA12* [46]. *IAA14* can interact with *ARF7*/*ARF19* to affect the rate of lateral root development [47]. The protein sequences of ARF and IAA from safflower were compared with the STRING database, revealing a protein–protein interaction network with 37 nodes and 216 edges (Figure 3). Eight members of the ARF family (*CtARF2*, *CtARF3*, *CtARF4*, *CtARF5*, *CtARF15*, *CtARF18*, *CtARF19*, and *CtARF22*) were identified within this interaction network, suggesting that they may be core members of the *CtARF* gene family involved in the regulation of safflower growth and development. Notably, all core members belonged to the ARF family, indicating that ARF family members are involved in a broader range of functions, which is consistent with the GO enrichment results of safflower. Transcriptome data analysis revealed significant differences in the expression of ARF genes among safflower with different flower colors. Additionally, it has been reported that *NtARF8* in tobacco can synergistically regulate vegetative growth and seed production with *TTG2*, and can control flower color by regulating anthocyanin biosynthesis genes [48]. Therefore, we speculate that the ARF family genes in safflower may also have similar functions.

### 3.4. The Response of Endogenous IAA and Active Ingredient Content in Safflower with Different Flower Colors to Exogenous Hormone Treatment

Flower color is one of the important traits of safflower. The expression patterns and active ingredient content of safflower with different flower colors showed significant differences (Figure 4). Studies have shown that the differences in auxin biosynthesis and transport pathways as well as ARF expression in safflower with different flower colors may affect flower color formation and maintenance [49,50]. It has also been found that *CtUGT9* may play an important role in the color transition of safflower and the biosynthesis of flavonoid glycosides [51]. In addition, our experiments found that the endogenous IAA, HSYA, and KAE content in safflower with different flower colors showed significant differences. The endogenous IAA and HSYA contents in RS were significantly higher than those in WS, while the KAE content was the opposite, which may be related to the growth environment and genetic background of the plants [50]. However, there is a lack of research evidence on whether auxin can regulate the content of flavonoids in safflower with different flower colors.

Plant growth regulators can influence plant growth by modulating the endogenous hormone levels within plants [52]. We treated safflower with different flower colors using various concentrations of plant growth regulators (IAA and NPA). The results showed that treatment with IAA75 and NPA50 significantly increased the IAA content in RS, while decreasing the IAA content in WS, with the effects of NPA treatment being more pronounced (Figure 6). This may be related to the influence of exogenous auxin on the processes of IAA biosynthesis and transport, as well as auxin signaling pathways [53]. Further research revealed that the application of exogenous plant hormones significantly increased plant height and had a pronounced effect on stem thickness (Figure 5). Auxin (IAA) promoted plant cell expansion and elongation, thereby increasing plant height and stem thickness. In contrast, NPA primarily interfered with the polar transport of auxin, thereby affecting plant growth. This effect is closely related to the concentration of exogenous application [54,55]. Furthermore, it has been found that high concentrations of NPA can inhibit the polar transport of auxin, leading to abnormal development of the hypocotyls in safflower [55]. The stigmas of safflower contain a variety of flavonoids, among which HSYA (hydroxysafflor yellow A) is the major bioactive compound. Stigmas with different flower colors have different contents of flavonoids [56]. Relatively speaking, RS contains more HSYA, while WS contains more KAE (Figure 8). It has been found that treatment with IAA75 significantly reduced the content of HSYA and increased the content of KAE. Treatment with NPA50 significantly increased the content of HSYA (in WS) and KAE (in RS), while decreasing the content of KAE in WS. This indicated that the content of active ingredients in safflower may be regulated by endogenous hormone levels. Furthermore, the results of real-time quantitative PCR showed that in RS, the expression levels of *ARF2*/*3*/*4*/*22* were in the order of NPA > IAA > CK (control), while the expression levels of *ARF15*/*19* were in the order of IAA > NPA > CK, with significant differences observed (Figure 7 and S5). However, in WS, the members of the ARF family showed different trends. The expression levels of *ARF2*/*3*/*4*/*5*/*15* gradually decreased, in the order of CK > IAA > NPA. In contrast, *ARF22* showed an opposite trend, with the highest expression level under NPA treatment. This suggested that *ARF2*/*3*/*4*/*5*/*15*/*19*/*22* respond to exogenous hormone treatment and are core members regulating the content of endogenous IAA and active ingredients.

## 4. Materials and Methods

### 4.1. Materials

In this study, two safflower cultivars, “Yumin spineless (red, RS)” and “Xin Honghua No. 7 (white, WS)”, were used as experimental materials. Safflower seeds were directly sown in pots (substrate composed of loam, vermiculite, and perlite in a volume ratio of 3:1:1), with 10 plants per pot and 3 pots per treatments, and placed outdoors for cultivation. When the safflower plants reached the jointing stage (with the stem elongating to approximately 2 cm), plants with inconsistent growth were removed (one safflower plant was retained per pot), and the treatments were initiated. Additionally, safflower plants grown in the experimental field were treated when they reached the blooming stage (with approximately 60% of the flowers open). The plants were treated with IAA and NPA (Appendix A), with applications occurring every five days a total of three times, ensuring even spraying. Fifteen days after treatment, root, stem, leaf, and flower samples were collected from the safflower plants, flash-frozen in liquid nitrogen, and stored at −80 °C for subsequent use.

All data obtained from the experiments were organized using Excel software. Analysis of variance (ANOVA) and graphing were performed using R (Version 4.4.1) and GraphPad Prism 9.5 software. The figures were refined and enhanced using Adobe Illustrator (Version 24.0.1) software.

### 4.2. Identification and Physicochemical Properties and Structural Analysis of Safflower ARF Transcription Factor Family Members

Our laboratory obtained the complete genome and annotation information of safflower. The protein sequences of all ARF members from *Arabidopsis thaliana* were downloaded from the TAIR (https://www.arabidopsis.org/, accessed on 18 October 2024) [57], and those from *Helianthus annuus* were obtained from the NCBI database (https://www.ncbi.nlm.nih.gov/, accessed on 27 October 2024) [58]. These sequences were aligned (e-value 1 × 10^−5^) against the safflower genome in our laboratory to identify homologous sequences, which were retained as candidate ARF family protein sequences. The Pfam (http://Pfam.sanger.ac.uk/, accessed on 20 November 2024) [59] and SMART (https://smart.embl.de/smart/set_mode.cgi?GENOMIC=1, accessed on 20 November 2024) [60] online analysis tools were used to annotate the domains of the candidate sequences. Sequences containing the PF06507 domain were ultimately identified as the candidate ARF sequences in safflower.

The basic physicochemical properties of ARF proteins were analyzed using the Prot Param online tool (https://web.expasy.org/protparam/, accessed on 28 November 2024) [61]. Subcellular localization was predicted using the Plant-mPLoc website (http://www.csbio.sjtu.edu.cn/bioinf/plant-multi/, accessed on 28 November 2024) [62]. The secondary structure of the proteins was predicted via the SPOMA website (https://npsa.lyon.inserm.fr/cgi-bin/npsa_automat.pl?page=/NPSA/npsa_sopma.html, accessed on 28 November 2024) [63]. The tertiary structure of the proteins was predicted using the SWISS-MODEL website (https://swissmodel.expasy.org/, accessed on 28 November 2024) [64].

### 4.3. Multiple Sequence Alignment and Phylogenetic Analysis

The ARF protein sequences obtained from safflower were aligned with those of *Arabidopsis thaliana* and *Helianthus annuus* using the ClustalW tool in MEGA7.0 [65] software. A phylogenetic tree was constructed using the neighbor-joining (NJ) method with a bootstrap value of 1000 (Appendix A). Finally, the tree was visually enhanced using the iTOL online tool (https://itol.embl.de/, accessed on 20 November 2024) [66].

### 4.4. Chromosomal Distribution, Gene Structure, and Collinearity Analysis

The chromosomal distribution map was constructed based on the chromosomal location information of the genes using TBtools (Version 2.210) [67]. The gene structure of the ARF family was visualized using the gff3 file in TBtools [67]. The gene duplication and collinearity analysis was completed using the Advanced Circos and Dual Synteny Plot MCscanX modules in TBtools [67].

### 4.5. Conserved Domain and Conserved Motif Analysis

The conserved domains of ARF proteins were analyzed using the online tool CD-search from NCBI (https://www.ncbi.nlm.nih.gov/Structure/cdd/wrps b.cgi, accessed on 23 November 2024) [68]. The conserved motifs of ARF proteins were analyzed using the online database MEME (https://meme-suite.org/meme/tools/meme, accessed on 23 November 2024) [69]. Finally, the results were visualized using TBtools [67].

### 4.6. Cis-Acting Element Analysis and Protein–Protein Interaction Prediction

The promoter sequences (2000 bp upstream of the start codon) of safflower ARF genes were extracted using TBtools (Appendix A) [67], and then input into the PlantCare website (https://bioinformatics.psb.ugent.be/webtools/plantcare/html/, accessed on 24 November 2024) [70] to predict cis-acting elements in the promoter regions. The distribution map of cis-acting elements and the heatmap were generated using TBtools [67]. The interactions between safflower ARF and IAA families were analyzed using the STRING website (https://cn.string-db.org/, accessed on 15 December 2024) [71]. The protein sequence analysis confidence was set to 0.4, and the output results were visualized using Cytoscape [72]. GO enrichment analysis was performed using TBtools [67].

### 4.7. Expression Pattern Analysis of ARF Genes in Different Flower Colors

Based on the safflower transcriptome database constructed by Wang et al. [30], the expression data of safflower with different flower colors (white—W, yellow—Y, light red—LR, and deep red—DR) were extracted. The expression patterns of the 23 ARF genes in different tissues were analyzed using TBtools [67] to generate a heatmap.

### 4.8. Determination of Endogenous IAA and Active Component Content

The endogenous IAA content in safflower was determined using a plant IAA content determination kit (Plant Indole-3-Acetic Acid (IAA) ELISA kit, Jingmei, Yancheng, China). The specific preparation method was referred to the product manual [73].

The content of active components (HSYA and KAE) in safflower flowers was determined by high-performance liquid chromatography (HPLC) [74]. The chromatographic column used was an Agilent EC-C-18 column (150 mm × 4.6 mm, 4 μm). The detection wavelengths were set at 403 nm and 367 nm. The mobile phase consisted of A: water, B: methanol, C: acetonitrile, and D: 0.4% phosphoric acid solution. The elution conditions were as follows: 0–60 min, 5% B and 95% D; 60–65 min, 95% B and 5% D; 65–70 min, 5% B and 95% D. The column temperature was maintained at 30 °C, with an injection volume of 10 μL. The standard concentrations were 0.424 mg/mL and 0.432 mg/mL.

### 4.9. RNA Extraction and Quantitative Real-Time Reverse Transcription PCR (qRT-PCR) Analysis

To detect the response of safflower ARF genes to exogenous hormones (IAA and NPA), roots, stems, leaves, and flowers of treated safflower were collected. The samples were ground in liquid nitrogen and RNA was extracted using the FastPure Plant Total RNA Isolation Kit (Vazyme, NanJing, China). The integrity of the extracted RNA was assessed by agarose gel electrophoresis, and its concentration was measured using a NanoDrop spectrophotometer (Thermo Fisher, Waltham, MA, USA). Subsequently, 0.5 μg of RNA was reverse transcribed into cDNA using the HiScript III RT SuperMix for qPCR (Vazyme). The diluted cDNA was used for quantitative real-time PCR to measure the expression levels of ARF genes. The qPCR was performed using the ChamQ SYBR Color qPCR Master Mix (Vazyme), with *Ct60s* as the reference gene [75]. The primers used are listed in Appendix A. Each sample was analyzed in triplicate, and the data were analyzed using the 2^−ΔΔCT^ method [76].

## 5. Conclusions

This study represents the first genome-wide analysis of ARFs in safflower. A total of 23 *CtARFs* were identified across the 12 chromosomes of safflower. These genes were analyzed in terms of conserved motifs, conserved domains, gene structure, secondary and tertiary protein structures, and phylogenetic evolution. These findings revealed the high conservation and evolutionary relationships among *CtARF* genes. Collinearity analysis revealed that tandem duplication is the primary driving force for the expansion of the *CtARF* gene family. The identification of cis-acting elements related to hormone response in the promoter regions of *CtARF* genes facilitated a better understanding of hormone response pathways in safflower. Through protein interaction network analysis, we identified the core members of the *CtARF* family (*CtARF2/3/4/5/15/18/19/22*) and studied their expression patterns in different organs of safflower under hormone treatments. Additionally, the study found that different hormone treatments led to significant changes in the endogenous auxin and bioactive compound contents of safflower with different flower colors, which may be related to the expression of ARF family members, especially the core members. The findings of this study will provide a reference for further research on the involvement of safflower ARF genes in plant hormone pathways and secondary metabolite synthesis pathways and will be of certain value for the breeding of safflower cultivars.

## Figures and Tables

**Figure 1 ijms-26-03773-f001:**
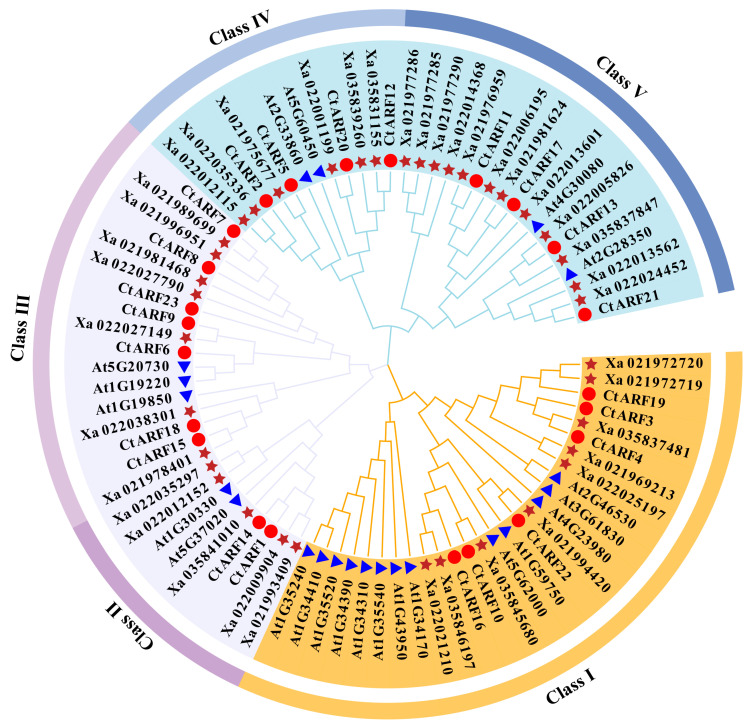
Phylogenetic relationships of ARF families in safflower, *Arabidopsis thaliana*, and *Helianthus annuus*. Colors represent different subfamilies, and shapes represent different species: circles for safflower, triangles for *Arabidopsis thaliana*, and stars for *Helianthus annuus*.

**Figure 2 ijms-26-03773-f002:**
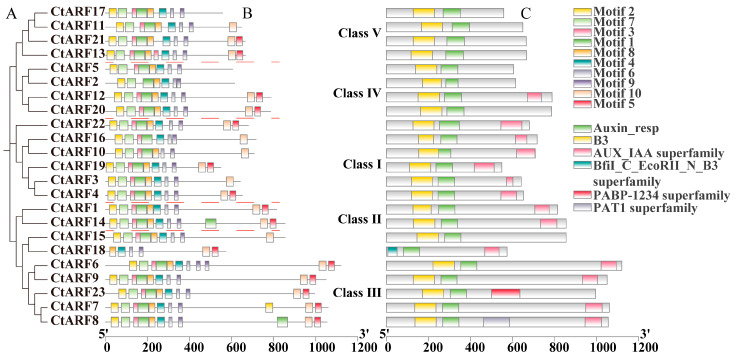
Analysis of conserved domains and conserved motifs in safflower ARF proteins. (**A**) Phylogenetic tree of the ARF gene family. (**B**) Conserved motifs of the ARF gene family. (**C**) Conserved domains of the ARF gene family.

**Figure 3 ijms-26-03773-f003:**
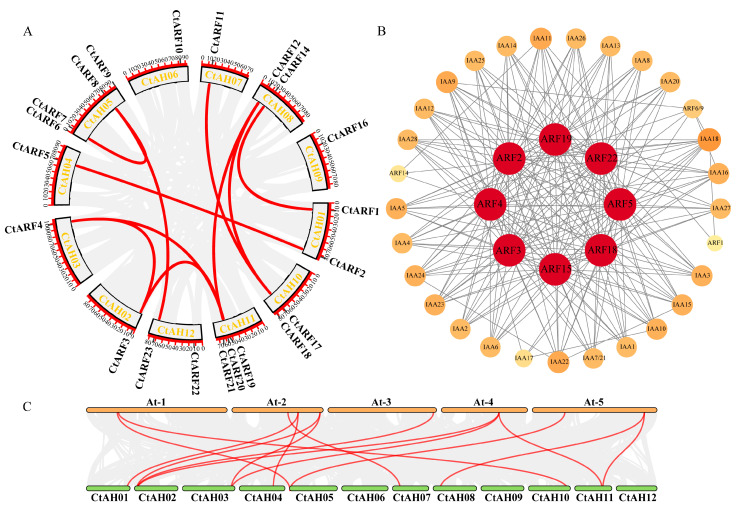
Intraspecific synteny analysis (**A**), interspecific synteny analysis (**C**), and protein–protein interaction analysis (**B**) of safflower ARF proteins.

**Figure 4 ijms-26-03773-f004:**
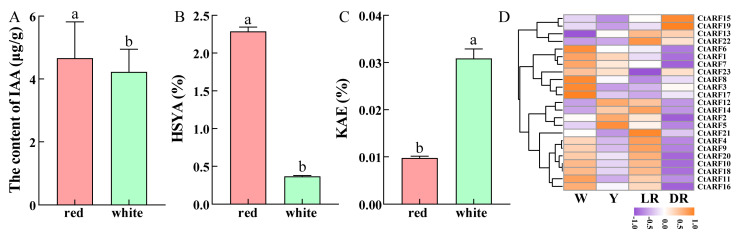
Content of endogenous IAA (**A**) and active components (HSYA (**B**) and KAE (**C**)) in safflower with different flower colors. HSYA represents hydroxysafflor yellow A, and KAE represents kaempferol. Different colors represent different safflower varieties, and different lowercase letters indicate significant differences. Expression analysis of the ARF gene family in safflower with different flower colors (**D**). W represents white safflower, Y represents yellow safflower, LR represents light red safflower, and DR represents dark red safflower.

**Figure 5 ijms-26-03773-f005:**
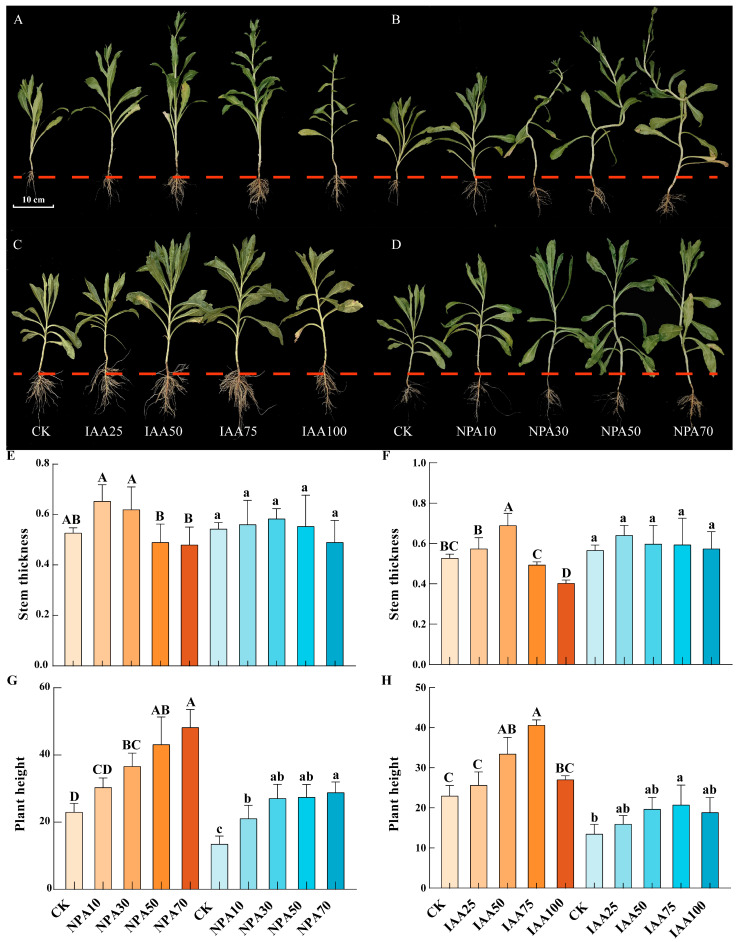
Growth status of safflower after exogenous hormone treatment (IAA and NPA). (**A**,**B**) Red safflower (RS) and (**C**,**D**) white safflower (WS). Measurement of agronomic traits of safflower after exogenous hormone treatment (IAA and NPA). (**E**,**G**) NPA treatment and (**F**,**H**) IAA treatment. Brown indicates red safflower (RS), and blue indicates white safflower (WS). The axes indicate the specific treatments, with CK representing the control group. Different letters represent significant differences (*p* < 0.05) among three plants, with three biological replicates each.

**Figure 6 ijms-26-03773-f006:**
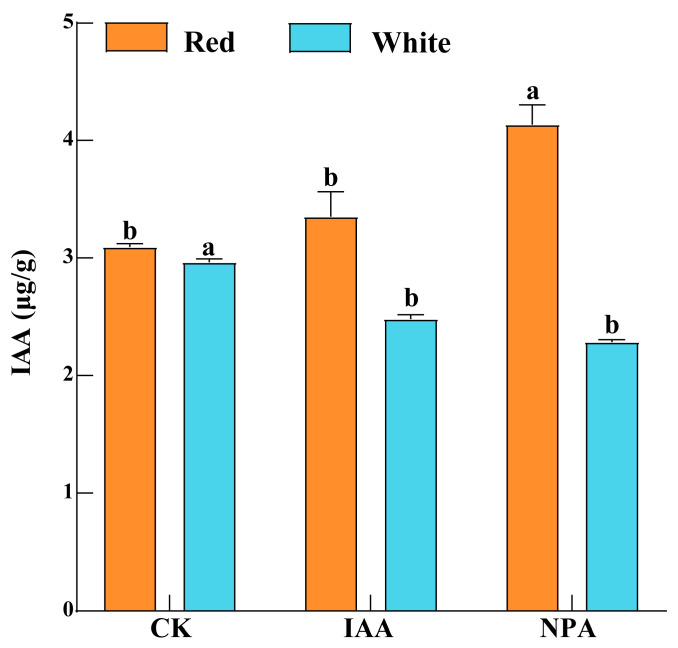
Determination of endogenous auxin content in safflower after exogenous hormone treatment (IAA and NPA). Brown represents red safflower (RS), and blue represents white safflower (WS). Different lowercase letters indicate significant differences (*p* < 0.05) among three plants, with three biological replicates each.

**Figure 7 ijms-26-03773-f007:**
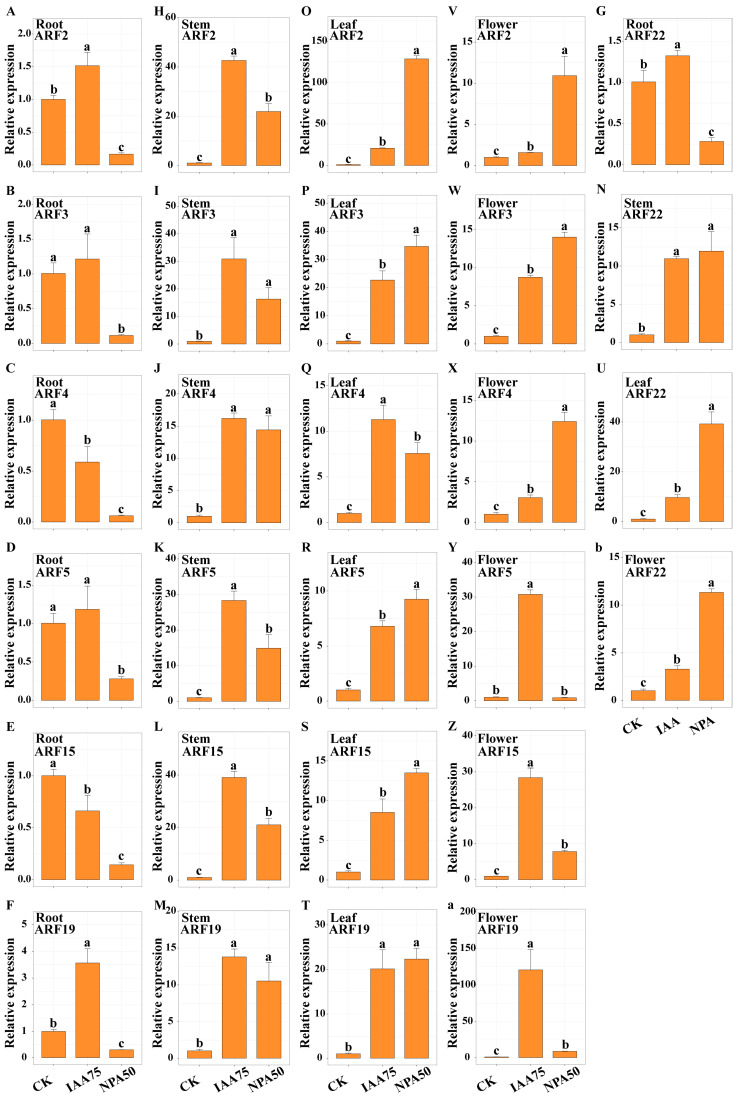
Determination of the relative expression levels of some ARF family members in different organs of safflower (RS) after exogenous hormone treatment (IAA and NPA). Root: **A/B/C/D/E/F/G**, Stem: **H/I/J/K/L/M/N**, Leaf: **O/P/Q/R/S/T/U**, Flower: **V/W/X/Y/Z/a/b**. Different lowercase letters indicate significant differences.

**Figure 8 ijms-26-03773-f008:**
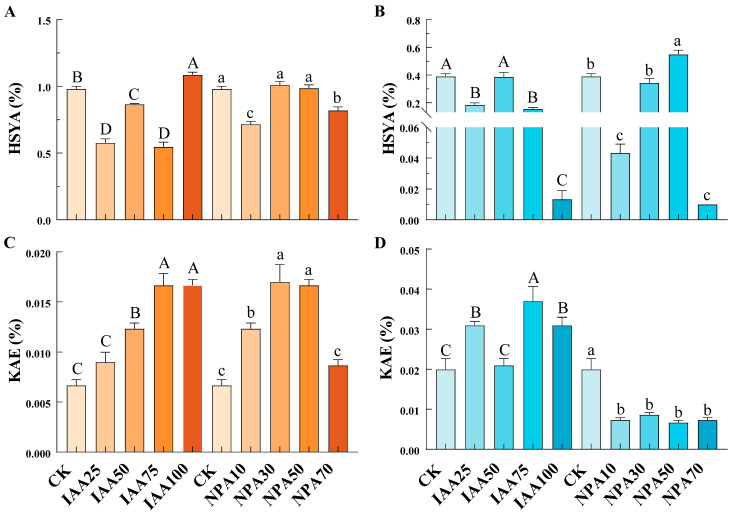
Determination of HSYA and KAE content in red safflower (RS; **A**,**C**) and white safflower (WS; **B**,**D**) after exogenous hormone treatment (IAA and NPA). Uppercase letters indicate IAA treatments, while lowercase letters indicate NPA treatments. Different letters represent significant differences (*p* < 0.05) among three plants, with three biological replicates each.

**Table 1 ijms-26-03773-t001:** Physiochemical properties of the *CtARF* gene family in safflower.

Gene Name	Locus ID	Protein Length (aa)	Molecular Weight (KDa)	PI	Instability Indices	Aliphatic Indices	Hydrophobicity	Subcellular Localization
*CtARF1*	CtAH01T0069600	813	90.53341	5.85	60.62	73.14	−0.468	nucleus
*CtARF2*	CtAH01T0211500	614	67.8474	8.38	53.06	71.04	−0.436	nucleus
*CtARF3*	CtAH02T0026700	641	71.89815	6.41	56.42	70.83	−0.582	nucleus
*CtARF4*	CtAH03T0268400	651	73.22105	5.51	55.27	73.01	−0.437	nucleus
*CtARF5*	CtAH04T0143600	604	67.71435	6.66	53.22	68.08	−0.483	nucleus
*CtARF6*	CtAH05T0001600	1119	125.48255	6.92	73.1	69.63	−0.638	nucleus
*CtARF7*	CtAH05T0047600	1059	117.82049	6.16	59.13	72.83	−0.551	nucleus
*CtARF8*	CtAH05T0213800	1054	116.50864	5.86	60.98	68.55	−0.575	nucleus
*CtARF9*	CtAH05T0279800	1049	117.9647	6.36	75.38	65.73	−0.765	nucleus
*CtARF10*	CtAH06T0272100	707	79.08847	6.08	68.44	63.41	−0.688	nucleus
*CtARF11*	CtAH07T0101600	648	72.28166	6.82	52.99	69.95	−0.44	nucleus
*CtARF12*	CtAH08T0107100	787	86.37022	5.94	56.73	70.32	−0.433	nucleus
*CtARF13*	CtAH08T0109300	665	73.11407	5.9	47.25	74.6	−0.362	nucleus
*CtARF14*	CtAH08T0176800	854	95.41481	5.81	63.57	70.2	−0.557	nucleus
*CtARF15*	CtAH08T0303600	855	94.24119	5.71	55.24	68.87	−0.521	nucleus
*CtARF16*	CtAH09T0132300	716	79.16385	6.84	58.87	65.21	−0.6	nucleus
*CtARF17*	CtAH10T0191300	557	62.06441	8.21	51.29	71.92	−0.447	nucleus
*CtARF18*	CtAH10T0242900	573	63.1186	5.13	57.16	71.61	−0.43	nucleus
*CtARF19*	CtAH11T0158500	548	61.86642	6.11	45.67	75.71	−0.451	nucleus
*CtARF20*	CtAH11T0163400	785	85.89191	6.06	54.01	69.87	−0.414	nucleus
*CtARF21*	CtAH11T0194100	664	72.94625	7.64	46.99	66.49	−0.462	nucleus
*CtARF22*	CtAH12T0040800	679	76.07008	5.75	57.36	76.91	−0.462	nucleus

## Data Availability

The original contributions presented in this study are included in the article/Appendix A. Further inquiries can be directed to the corresponding authors.

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
