# Peer review of "Genome-Wide Identification of the ARF Gene Family in Safflower (Carthamus tinctorius L.) and Their Response Patterns to Exogenous Hormone Treatments"

_ijms, 2025, doi:10.3390/ijms26083773_

Round 1

Reviewer 1 Report

Comments and Suggestions for Authors

The manuscript “Genome-wide identification of the ARF Gene Family in safflower (Carthamus tinctorius L.) and functional analysis after exogenous hormone treatment” presents a comprehensive genome-wide identification and characterization of the ARF gene family in safflower (Carthamus tinctorius L.). The study is generally well-prepared and provides valuable insights into the role of ARF genes in safflower growth, development, and active component biosynthesis. However, some revisions are required to strengthen the manuscript.

  1. Grammatical Errors and Phrasing: There are occasional grammatical errors and awkward phrasing throughout the manuscript. For example: Line 14: "ARFs" should not be italicized. Lines 189 and 555: The phrase "evenly distributed" is not appropriate in these contexts. Lines 262-263: The phrase "showing an opposite trend" is unclear and needs clarification. Some abbreviations (e.g., HSYA, KAE) are not defined upon first use.
  2. The title mentions "functional analysis," but the manuscript lacks substantial functional validation (e.g., gene knockout or overexpression). Provide functional validation if it is possible in safflower. Alternatively, the title should be revised.
  3. Lines 101-102 mention the identification of 89 homologous genes in the safflower genome, but later, only 23 ARF genes are reported. Please clarify this discrepancy and confirm the correct number of ARF genes identified.
  4. The manuscript contains an excessive number of figures, some of which could be moved to supplementary materials to improve readability. For example, Figures 3, 4, and 8 are more suitable as supplementary figures.
  5. Lines 241-254: The results section should include more details about the analysis methods and data sources (e.g., transcriptomic data or qRT-PCR). Additionally, the specific tissues used for expression analysis should be clearly stated. The Materials and Methods section mentions expression analysis in different tissues, but this is not reflected in the results or figures.
  6. The manuscript should avoid overemphasizing flower color, as the study does not specifically focus on this trait. Statements like "The expression levels of CtARF1/6/7/8/16/17 gradually decreased with the deepening of flower color (W-Y-LR-DR), while the expression levels of CtARF13/19 gradually increased with the deepening of flower color" should be removed, as the small sample size does not support a correlation between ARF genes and flower color.
  7. Lines 255-263: The rationale for measuring endogenous hormones and active components is unclear, and no clear connection is established between them.
  8. Figure 13: The measurement of endogenous hormone content after hormone treatment needs further explanation. Why do different safflower lines show opposite changes in endogenous hormone levels after treatment? This should be discussed in more detail.
  9. Figures 14 and 15 should be combined into a single figure to improve clarity. Additionally, the x-axis labels in these figures must be clearly defined and should not be omitted or arbitrarily named. For example, labels like "RCK" and "WCK" are too vague and should be replaced with more formal terms. Figure 16 should also be revised accordingly to ensure consistency and clarity in labeling.
  10. The manuscript states that samples were collected 15 days after hormone treatment. Please clarify whether this time point was chosen based on preliminary data or prior literature.

Author Response

Comments 1: Grammatical Errors and Phrasing: There are occasional grammatical errors and awkward phrasing throughout the manuscript. For example: Line 14: "ARFs" should not be italicized. Lines 189 and 555: The phrase "evenly distributed" is not appropriate in these contexts. Lines 262-263: The phrase "showing an opposite trend" is unclear and needs clarification. Some abbreviations (e.g., HSYA, KAE) are not defined upon first use.
Response 1: Thank you for pointing this out. We agree with this comment. Therefore, I have made the following revisions.

1:Line 14: "ARFs" should not be italicized. 

Revised.

2:Lines 189 and 555: The phrase "evenly distributed" is not appropriate in these contexts. Revised. The original sentence has been revised to “are distributed across all chromosomes except chromosome 6” and ”The term “evenly” has been removed”.

3:Lines 262-263: The phrase "showing an opposite trend" is unclear and needs clarification. Revised. The original sentence has been supplemented with a clear description “RS contains a higher amount of HSYA, while WS contains a higher amount of KAE”

4:Some abbreviations (e.g., HSYA, KAE) are not defined upon first use. 

Revised. The abbreviations have been defined “Hydroxysafflor yellow A (HSYA)” and “KAE (Kaempferol)”.
Comments 2: The title mentions "functional analysis," but the manuscript lacks substantial functional validation (e.g., gene knockout or overexpression). Provide functional validation if it is possible in safflower. Alternatively, the title should be revised.
Response 2: Agree. I have revised the article title to “Genome-wide identification of the ARF Gene Family in safflower (Carthamus tinctorius L.) and their response patterns to exogenous hormone treatments.”
Comments 3: Lines 101-102 mention the identification of 89 homologous genes in the safflower genome, but later, only 23 ARF genes are reported. Please clarify this discrepancy and confirm the correct number of ARF genes identified.
Response 3: Thank you for the suggestion. A total of 89 homologous genes were initially identified as candidate sequences. Subsequently, these candidates were verified through the Pfam database, ultimately resulting in the identification of 23 ARF genes. I have revised the wording, adding the transitional phrase “In summary” and an explanation “preliminary screening”.
Comments 4: The manuscript contains an excessive number of figures, some of which could be moved to supplementary materials to improve readability. For example, Figures 3, 4, and 8 are more suitable as supplementary figures.
Response 4: Thank you for your suggestions. Figures 1,3, 4, 7, 8, and 15 have been moved to the supplementary materials.
Comments 5: Lines 241-254: The results section should include more details about the analysis methods and data sources (e.g., transcriptomic data or qRT-PCR). Additionally, the specific tissues used for expression analysis should be clearly stated. The Materials and Methods section mentions expression analysis in different tissues, but this is not reflected in the results or figures.
Response 5: Thank you for your comments. The manuscript has been revised accordingly. The data source and analysis methods have been specified, and the tissue used for expression analysis has been clearly defined. “The transcriptome data were derived from the safflower (Carthamus tinctorius L.) transcriptome database constructed by Wang et al., with sequencing materials obtained from safflower inflores-cences. Subsequently, the transcriptome data were analyzed using the “pheatmap” package in R to generate expression heatmaps”.
Comments 6: The manuscript should avoid overemphasizing flower color, as the study does not specifically focus on this trait. Statements like "The expression levels of CtARF1/6/7/8/16/17 gradually decreased with the deepening of flower color (W-Y-LR-DR), while the expression levels of CtARF13/19 gradually increased with the deepening of flower color" should be removed, as the small sample size does not support a correlation between ARF genes and flower color.
Response 6: Agree. I have removed this sentence.
Comments 7: Lines 255-263: The rationale for measuring endogenous hormones and active components is unclear, and no clear connection is established between them.
Response 7: Thank you for your comments. Indeed, there is no clear relationship between them. In this section, I aim to demonstrate that there are significant differences in endogenous auxin and active compound content between the untreated RS and WS groups. In other words, there are differences in metabolite content between different safflower varieties (different flower colors).
Comments 8: Figure 13: The measurement of endogenous hormone content after hormone treatment needs further explanation. Why do different safflower lines show opposite changes in endogenous hormone levels after treatment? This should be discussed in more detail.
Response 8: Thank you for your comments. I have revised the manuscript accordingly, adding more detailed descriptions and discussing the potential causes of the differences. “This variation may be due to different responses of different safflower varieties to exogenous hormones. The auxin signaling pathway or auxin biosynthesis-related genes in RS may be more sensitive to exogenous IAA and NPA, thereby being activated and promoting the synthesis of endogenous auxin. In contrast, WS may be more sensitive to feedback regulation, and exogenous IAA treatment may inhibit the synthesis of endogenous hormones. Additionally, this phenomenon may also be related to differences in gene expression, as well as the physiological state and genetic background of the plants”.
Comments 9: Figures 14 and 15 should be combined into a single figure to improve clarity. Additionally, the x-axis labels in these figures must be clearly defined and should not be omitted or arbitrarily named. For example, labels like "RCK" and "WCK" are too vague and should be replaced with more formal terms. Figure 16 should also be revised accordingly to ensure consistency and clarity in labeling.
Response 9: Thank you for your comments. I have made the revisions accordingly. I have changed the X-axis labels of Figures 14, 15, and 16 to match the treatment labels. The clarity of these figures has been improved. Given the large number of results from qRT-PCR, Figure 15 has been moved to the supplementary materials.
Comments 10: The manuscript states that samples were collected 15 days after hormone treatment. Please clarify whether this time point was chosen based on preliminary data or prior literature.
Response 10: Thank you for your comments. The selection of this time point was based on our laboratory's long-term experiments. We had previously conducted preliminary experiments on hormone treatments and found that this time point yielded optimal results.

Reviewer 2 Report

Comments and Suggestions for Authors

This manuscript focused on the bio-informatic analysis of ARF gene family in safflower, and combined the qRT-PCR experiments to make contributions for investigation into the involvement of safflower. Generally, we believe this manuscript are of innovation and significance for safflower studies, while many technical issues must be put forward first.

  • The writing of the manuscript needs improvement, and suggest to invite one professional researcher to polish this article.
  • Line 35, suggest to revise Safflower (Carthamus tinctorius L) as Safflower (Carthamus tinctorius L.).
  • Figure 1, suggest to put the CtAH01-12 in the bottom of the relative chromosome.
  • Figure 2, suggest to revise the gene names as the At, Xa, and Ct。
  • Too many figures were presented in the manuscript, and normally less than 8 ones are accept. Suggest to combine or remove some ones into supplemental file.
  • Normally, figure 3 and figure 5 were combined to present, while figure 4 could be removed into supplemental file.
  • Figure 9, please add the notes for explaining the W, Y, LR, and DR.
Comments on the Quality of English Language

The writing of the manuscript needs improvement, and suggest to invite one professional researcher to polish this article.

Author Response

Comments 1: The writing of the manuscript needs improvement, and suggest to invite one professional researcher to polish this article.

Response 1: Thank you for pointing this out. We have further polished the manuscript.

Comments 2: Line 35, suggest to revise Safflower (Carthamus tinctorius L) as Safflower (Carthamus tinctorius L.).
Response 2: Agree. Revisions have been made accordingly.

Comments 3: Figure 1, suggest to put the CtAH01-12 in the bottom of the relative chromosome.
Response 3: Agree. Revisions have been made accordingly.

Comments 4: Figure 2, suggest to revise the gene names as the At, Xa, and Ct.
Response 4: Agree. Revisions have been made accordingly.

Comments 5: Too many figures were presented in the manuscript, and normally less than 8 ones are accept. Suggest to combine or remove some ones into supplemental file.

Response 5: Agree. Figures 1,3, 4, 7, 8, and 15 have been moved to the Supplementary Materials.

Comments 6: Normally, figure 3 and figure 5 were combined to present, while figure 4 could be removed into supplemental file.
Response 6: Agree. Revisions have been made accordingly.

Comments 7: Figure 9, please add the notes for explaining the W, Y, LR, and DR.
Response 7: Agree. Figure notes have been added to explain W, Y, LR, and DR. “W represents white safflower, Y represents yellow safflower, LR represents light red safflower, and DR represents dark red safflower”.

4. Response to Comments on the Quality of English Language
Point 1:The writing of the manuscript needs improvement, and suggest to invite one professional researcher to polish this article.
Response 1: Thank you for pointing this out. We have further polished the manuscript.

Reviewer 3 Report

Comments and Suggestions for Authors

Comments and Suggestions for Authors

This manuscript analyzed the ARF gene family in safflower by conducting a comprehensive comparison analysis. This study shows well-planned experimental results, and these results are thought to be of interest to readers. I believe the manuscript is suitable for publication in the IJMS after some revisions.

Concerns:

  1. The conclusion section needs to be followed by discussion.
  2. Some Figures are suggested arrange to the supplementally Figures, such as Fig. 1, Fig. 4, Fig. 7. Some Figs need to be merged, such as Fig. 3 and 5, Fig. 9 and 10, Fig. 11, 12 and 13. Figures 14 and 15 need to be merged and reassembled, the author can arrange different genes expression patterns at same tissue in one bar chart.
  3. Table S2 contains important information and is recommended as the main table, while Table 1 can be placed in the supplementally table.
  4. Fig. 9, The abbreviations of the letters W, Y, LR, DR, need to be marked, as well in the following Figures.
  5. Fig. 10A, I don’t trust the difference between red and white due to the error bars.
  6. Statistical methods need to be detailed in the MM section, and the significance markers need to be indicated in all the corresponding figures.
  7. Figures should be cited in Discussion section.
  8. In reference, Journal full name is mixed with abbreviation, meanwhile, only the first word of the title needs to be capitalized (ref9, ref9, etc), please checked.
Comments on the Quality of English Language

The English can be improved.

Author Response

Comments 1: The conclusion section needs to be followed by discussion.

Response 1:Thank you for your comments. I have added some discussion after the Conclusions section.
Comments 2: Some Figures are suggested arrange to the supplementally Figures, such as Fig. 1, Fig. 4, Fig. 7. Some Figs need to be merged, such as Fig. 3 and 5, Fig. 9 and 10, Fig. 11, 12 and 13. Figures 14 and 15 need to be merged and reassembled, the author can arrange different genes expression patterns at same tissue in one bar chart.
Response 2: Agree. Figures 1, 3, 4, 7, 8, and 15 have been moved to the Supplementary Materials, and Figures 9 and 10 have been merged, as well as Figures 11 and 12.
Comments 3: Table S2 contains important information and is recommended as the main table, while Table 1 can be placed in the supplementally table.
Response 3: Agree. Revisions have been made accordingly.
Comments 4: Fig. 9, The abbreviations of the letters W, Y, LR, DR, need to be marked, as well in the following Figures.
Response 4: Agree. Figure notes have been added to explain W, Y, LR, and DR. “W represents white safflower, Y represents yellow safflower, LR represents light red safflower, and DR represents dark red safflower” and “HSYA represents Hydroxysafflor yellow A, and KAE represents Kaempferol”.
Comments 5: Fig. 10A, I don’t trust the difference between red and white due to the error bars.
Response 5: Agree. The differences are indeed relatively small due to the error bars. However, analysis using R software indicates that the differences are statistically significant (P < 0.05).
Comments 6: Statistical methods need to be detailed in the MM section, and the significance markers need to be indicated in all the corresponding figures.
Response 6: Agree. I have added a description of the statistical methods in the Materials and Methods section and have indicated the significance markers at the corresponding positions in the figures.
Comments 7: Figures should be cited in Discussion section.
Response 7: Agree. Revisions have been made accordingly.
Comments 7: In reference, Journal full name is mixed with abbreviation, meanwhile, only the first word of the title needs to be capitalized (ref9, ref9, etc), please checked.
Response 7: Agree. Revisions have been made accordingly.
4. Response to Comments on the Quality of English Language
Point 1:The writing of the manuscript needs improvement, and suggest to invite one professional researcher to polish this article.
Response 1: Thank you for pointing this out. We have further polished the manuscript.

Reviewer 4 Report

Comments and Suggestions for Authors

The paper extends to another plant the analysis of ARF  family members

the authors specialize in examining links to flower color for safflower      the  choice of HSYA   assessment seems sound but do not understand KEA  also used   -  see sticky notes 

A major request for revision is to eliminate all the vague references to hormones   -  the text even in abstract should delineate the molecules involved    AND  point out that NPA  is a man-made product   would not normally be in a plant tissue     And is this a suitable treatment as it is thought to inhibit transport rather than synthesis  -  so confusion in my mind  

the methods require more detail  in several places

the discussion section suddenly shifts to crocus  sativum  -  not sure why  or  was this a typo  or ? 

unclear of the replication of this study -  how many plants /treament  and how many complete replicates

not sure why data is shown with certain units    -   please review 

the images of the plants do not always seem to agree with the description of what happened to the plants

there is no overall conclusion with meaning  from the factual study 

Author Response

Comments 1: The question about the sticky note in the abstract section is answered as follows.
Response 1:Thank you for pointing this out. We agree with this comment. 

1:do not understand use of term “comprehensive economic” OK. 

Revised. delete ”comprehensive”

2:what is significance of B3 domain?

Revised. The B3 domain is a unique structural domain of the ARF family. Our analysis was informed by previous studies on the ARF family in other plant species.

3:are the numbers of these genes assigned randomly by you or what was the rationale?

We designated the members of the CtARF family based on their chromosomal locations.

4:what is CtARF ? not explained

Ct denotes the abbreviation of the Latin name for Carthamus tinctorius.

5:think its better to state in flowers of all colors. this presumably means petals?

Revised. We added the expression analysis of different genes across various flower colors in safflower. The materials used for transcriptome sequencing were inflorescences of safflower.

6:what hormones? seems out of place no justification if more than IAA

Revised. Analysis of cis-acting elements revealed that members of the ARF family contain IAA-responsive elements (AuxRR-core).

7:what is this? you have not stated if up or down regulated

After treating safflower with exogenous hormones, we examined the relative expression levels of ARF family members via qRT-PCR and found that their expression levels indeed underwent significant changes. Additionally, analysis of cis-acting elements revealed that ARF family members contain auxin-responsive elements (AuxRR-core). These findings suggest a potential association between the observed changes in expression and the presence of auxin-responsive elements within the ARF family.
Comments 2: The question about the sticky note in the introduction section is answered as follows.
Response 2: Thank you for pointing this out. We agree with this comment. 

1:please write about global connections its grown also outside of China

Revised. We have incorporated a description of the global cultivation regions of safflower. “Moreover, safflower is cultivated in numerous regions across the globe, including countries such as the United States, India, and Canada.”

2:what part ? petals stigma whole flower etc

Revised. The primary medicinal part of safflower is its inflorescence.

3:but mainly thought of as IAA. is this your definition too? not stated

Revised. We have included relevant references.

4:root growth too

Revised. 

5:IAA. please replace auxin with IAA if that is what is meant. auxin loosely was used for any hormone in the past

Revised. 

6:please provide diagram to aid readers

The referenced literature provides detailed explanations, including graphical illustrations.

7:it is known how many genes are in the pathway ? ie how many and which genes may be controlled

Revised. We have added a description of the HSYA biosynthetic pathway. “Studies have indicated that genes such as cinnamate-4-hydroxylase (C4H), chalcone synthase (CHS), C-glycosyltransferase (CGT), and cytochrome P450 (P450) may be in-volved in the biosynthetic pathway of HSYA”.
Comments 3: The question about the sticky note in the results section is answered as follows.
Response 3: Thank you for pointing this out. We agree with this comment. 

1:UTR? CDS?

Revised. 

2:again for me this is vague. is it IAA or ethylene or ABA or SA or JA etc

Revised. We have provided detailed descriptions. “Both TGA-element and AuxRR-core are iaa-responsive elements”.

3:what traits does flower color influence? pollination or insect visitation? resilience to UV? to heat? to drought stress?

Revised. We have provided detailed descriptions. “Flower color, as a significant trait of safflower, not only provides ornamental value but also attracts pollinating insects”.

4:what do you mean active components?

Revised. The term has been revised to ”active compounds”.

5:do white colored flowers still contain the genes for pigmentation but in an inactivated form?

White flowers possess pigment-related genes, and the occurrence of white flowers is regulated by multiple factors, likely associated with both environmental and genetic contexts.

6:you show only two red white is this dark red?

There are numerous cultivars of safflower, and the one we selected is the "Yumin spineless" cultivar, which is primarily cultivated in Xinjiang, rather than the dark red type. The transcriptomic data are derived from Wang et al.

7:OK. please clarify in abstract that IAA was a hormone used. as was NPA

Revised. This has been clarified in the abstract.

8:so white can respond to exog IAA. but red does not?

Both A and B are safflower, while C and D are white flowers. A and C were treated with exogenous IAA (significant elongation was observed in both safflower and white flowers), whereas B and D were treated with exogenous NPA (severe deformity was observed in safflower, and a milder degree of deformity was observed in white flowers). 

9:what tiny roots! is this normal? what were growth conditions

The growth conditions consisted of a substrate mixture of nutrient soil: vermiculite: perlite at a volumetric ratio of 3:1:1. We watered thoroughly each time and placed the plants outdoors. The short root lengths may have resulted from accidental damage during extraction, causing partial root detachment. Additionally, the cleaning of the roots during later photographic documentation may have also led to some root loss.

10:these results do not look like images. is everything labelled correctly

These are the results of our experiments. Panels A and B show stem diameter, while panels C and D show plant height. Panels A and C represent NPA treatment, and panels B and D represent IAA treatment. Additionally, different colors of the bars represent different flower colors of safflower, with brown indicating red flowers and blue indicating white flowers.

11:please explain all the x axis labels. the S seems to be missing

Revised. We have provided detailed explanations.

12:but what tissues please confirm in text here

Revised. We have confirmed the tissues.

13:this was foliar spray. not root applied?

We employed foliar spraying as the application method.

14:this first sentence does not tell what is being looked at. if HYSA is yellow why is it found in both. need to understand the pathways of intermediates in the safflower petal color

Revised. We have provided detailed explanations.

15:again back to vague hormone use. I was expecting to see ABA or JA because the corresponding elements were detected in the genes but again it is only IAA or NPA. please clarify as to not confuse the reader

IAA and NPA were the exogenous hormones used in our treatments. In this section, we aim to demonstrate the changes in endogenous hormone levels (we measured endogenous IAA content), gene expression levels (we assessed the expression of key ARF family members), and bioactive compound contents (we quantified the levels of HSYA and KAE) following different hormone treatments (IAA is an auxin, and NPA is an auxin transport inhibitor).

16:giving only relative levels means that we do not know. how these different organs compare for expression of the different genes. how do the different organs compare?

In this section, we aim to clarify the expression profiles of core ARF family members in different organs of safflower following exogenous hormone treatments. We are investigating whether ARF family members in different organs also respond to exogenous hormone treatments. Comparisons of the expression of the same ARF genes across different organs will be conducted in our subsequent studies.

17:do not understand what the % term means need more detail

The content of the bioactive compounds was referenced from the Chinese Pharmacopoeia, which provides explicit definitions for the contents of HSYA and KAE. The “%” indicates the percentage content of HSYA and KAE in the dried inflorescence of safflower.

18:how does KAE relate to HSYA?

There is no direct correlation between the two. However, in our previous research, we found that red safflower flowers contain higher levels of HSYA and lower levels of KAE, while white safflower flowers have higher levels of KAE and lower levels of HSYA. We aim to utilize these two distinctly contrasting bioactive compounds to substantiate our research conclusions.

19:muddled. why is Crocus sativus introduced?

Revised. The term has been revised to “safflower”.

20:please comment somewhere that IAA is the native product whereas NPA is a made by chemistry commercially !

Revised. 

21:again how are KAE and HSYA related?

There is no specific correlation between the two, as referenced in response 18.
Comments 4: The question about the sticky note in the Materials and Methods section is answered as follows.
Response 4: Thank you for pointing this out. We agree with this comment. 

1:how did this alter the numbers of replicates

Revised. We have provided detailed descriptions. We thinned out the plants with inconsistent growth, retaining one plant per pot for treatment.

2:so what is the red in all your studies LR or DR?

Revised. There are numerous cultivars of safflower. The red safflower we used is the "Yumin spineless" cultivar, which is the predominant cultivar in Xinjiang, rather than the other cultivars mentioned in the transcriptome sequencing.
Comments 5: The question about the sticky note in the conclusions section is answered as follows.
Response 5: Thank you for pointing this out. We agree with this comment. 

1:at the moment only fact level

Revised. We have rephrased the conclusion section.

Round 2

Reviewer 3 Report

Comments and Suggestions for Authors

It's well revised.

Author Response

Comments 1: Is the research design appropriate?
Response 1: Thank you for pointing this out. Initially, we set up three pots per treatment, with 10 safflower plants in each pot. However, due to factors such as seed quality, the growth of the plants might be inconsistent, which could affect the subsequent experiments. Therefore, we removed the safflower plants with inconsistent growth, retaining only one plant per pot (since there were three pots per treatment, this resulted in three replicates per treatment) for the subsequent hormone treatments.

Reviewer 4 Report

Comments and Suggestions for Authors

the authors partially revised the manuscript  only. 

There are many places where the wording is poor

The methods lack adequate description of  the replication 

There is still frequent references to crocus

the connections between  HYSA  and KAE  is not well explained    Crocus work comes up here   do not understand

Comments on the Quality of English Language

the work requires editing for correct scientific wording 

Author Response

Comments 1: upon treatment with hormones that affect IAA content.
Response 1: Revised. 
Comments 2: The question about the sticky note in the introduction section is answered as follows.
Response 2: Agree. 

1:what is its structure 

chalcone?

how much of this would overlap KAE synthesis pathway?

ie a blockage after KAE synthesis would stop HYSA synthesis and this might account for white v yellow flowers?

Revised. Its structure is C-glucosylquinochalcone. The biosynthetic pathways of kaempferol (KAE) and hydroxysafflor yellow A (HSYA) indeed intersect, with the biosynthetic route of naringenin chalcone being conserved (catalyzed by enzymes such as cinnamate-4-hydroxylase (C4H) and chalcone synthase (CHS)).

2:what sort of compound is HYSA

becomes important since you look at KAE v  HYSA  later

Revised. 

3:(NPA) is a chemical that alters IAA transport and thus affect the responses to indigenous IAA in planta.

Revised.

4:encoding

Revised.

5:are they also involved in KAE is KAE a precursor of HSYA

please add this discussion

Revised. add “Additionally, Kaempferol (KAE) is synthesized through the catalytic actions of enzymes such as cinnamate-4-Hydroxylase (C4H), chalcone synthase (CHS), chalcone isomerase (CHI), and flavanone 3-hydroxylase (F3H). The biosynthetic pathways of KAE and HSYA indeed intersect, with the biosynthetic route of naringenin chalcone being conserved”.
Comments 3: The question about the sticky note in the results section is answered as follows.
Response 3: Agree. 

1:preliminary screen of the  safflower genome

Revised. 

2:are

Revised. 

3:actually its the gene products that play a role the genes do not

Revised. 

4:so wording could be the ARF proteins likely play important roles in

Revised. 

5:IAA-related events

Revised. 

6:you should also comment that the yellow pigment has medicinal value  

as in Intro  again

this seems a huge reason to look at these ARFs

Revised. The yellow pigments have been specifically described.

7:is there an actual published reference

add this reference if there is

if not is is Wang et al unpublished

Revised. add reference

8:two antioxidant

Revised.

9:no cap  k

Revised.

10:There were...

Revised.

11:you bracketed  words  are repetitive  so removed

Revised.

12:please add a sentence about KAE 

why do you mention this   

my understanding is although it has a different structure than HSYA  it also is an antioxidant  and as such it could be of medicinal use

Revised. add “KAE is a flavonoid compound, whose biosynthetic pathway partially overlaps with that of HSYA, and it exhibits pharmacological functions such as anticancer activity and antioxidant capacity.”

13:where was the deformity  seen    

ie please describe in words

Revised. “Specific manifestations include stem bending and leaf curling. The degree of malformation in RS is more severe, with a more pronounced degree of bending.”

12:change wording to 

exogenous IAA and an inhibitor of IAA transport, NPA. 

NPA is not a hormone    it inhibits a function of the hormone   here its transport

Revised. 

13:ia this light or dark red?

Revised. The cultivar we used is "Yumin spineless," which is a major cultivated variety in Xinjiang and is characterized by its deep red color.

14:axes indicate

Revised. 

15:at what P value  and what is  the number of plants and how many replicates

Revised. 

16:was highest when treated with NPA and then endogenous IAA.

Revised. 

17:IAA and NPA

Revised. 

18:treatments with IAA  or NPA.

Revised. 

19:IAA

Revised. 

20:in the plants treated with

Revised. 

21:treatment

Revised. 

22:it is not correct with errors to use two decimal places  round off

We have referred to other articles published in MDPI and found that they follow the same practice. Could you please provide a clear method for correction?

23:must give more  detail  P  value  and  the information on numbers of plants and full replications

Revised. 
Comments 4: The question about the sticky note in the discussion section is answered as follows.
Response 4: Agree. 

1:really crocus again?

Revised. All changes have been completed.

2:crocus?

Revised.

3:i am confused by crocus  again     

is thisother work

Revised.
Comments 5: The question about the sticky note in the Materials and Methods section is answered as follows.
Response 5: Agree. 

1:what is the gradient?

Revised. The term "gradient" has been changed to "treatment".

2:I do not understand

how many plants per treatment were used

how many times was the total study repeated

Revised. Initially, we set up three pots per treatment, with 10 safflower plants in each pot. However, due to factors such as seed quality, the growth of the plants might be inconsistent, which could affect the subsequent experiments. Therefore, we removed the safflower plants with inconsistent growth, retaining only one plant per pot (since there were three pots per treatment, this resulted in three replicates per treatment) for the subsequent hormone treatments.
 4. Response to Comments on the Quality of English Language
Point 1:the work requires editing for correct scientific wording 
Response 1: Thank you for pointing this out. We have further polished the manuscript.

Round 3

Reviewer 4 Report

Comments and Suggestions for Authors

improving    but data based on extracts from three plants only insufficient

Comments on the Quality of English Language

many sentences are  clumsy    -excessive wording makes reading hard